# Relationship of Mediterranean Diet and Its Components with Parameters of Structure, Vascular Function, and Vascular Aging in Subjects Diagnosed with Long COVID: BioICOPER Study

**DOI:** 10.3390/nu17203226

**Published:** 2025-10-14

**Authors:** Alicia Navarro-Cáceres, Leticia Gómez-Sánchez, Silvia Arroyo-Romero, Nuria Suárez-Moreno, Andrea Domínguez-Martín, Cristina Lugones-Sánchez, Susana González-Sánchez, Emiliano Rodríguez-Sánchez, Luis García-Ortiz, Marta Gómez-Sánchez, Elena Navarro-Matias, Manuel A. Gómez-Marcos

**Affiliations:** 1Primary Care Research Unit of Salamanca (APISAL), Salamanca Primary Care Management, Institute of Biomedical Research of Salamanca (IBSAL), 37005 Salamanca, Spain; alicia.nav@usal.es (A.N.-C.); silvia_ar@usal.es (S.A.-R.); nuria.suarez@usal.es (N.S.-M.); andreadm@usal.es (A.D.-M.); crislugsa@gmail.com (C.L.-S.); gongar04@gmail.com (S.G.-S.); emiliano@usal.es (E.R.-S.); lgarciao@usal.es (L.G.-O.); enavarro@saludcastillayleon.es (E.N.-M.); 2Castilla and León Health Service-SACYL, Regional Health Management, 37005 Salamanca, Spain; 3Emergency Service, University Hospital of La Paz, P. of Castellana, 261, 28046 Madrid, Spain; leticiagmzsnchz@gmail.com; 4Research Network on Chronicity, Primary Care and Health Promotion (RICAPPS), 37005 Salamanca, Spain; 5Department of Medicine, University of Salamanca, 28046 Salamanca, Spain; 6Department of Biomedical and Diagnostic Sciences, University of Salamanca, 37007 Salamanca, Spain; 7Home Hospitalization Service, Marques of Valdecilla University Hospital, s/n, 39008 Santander, Spain; martagmzsnchz@gmail.com

**Keywords:** long COVID, Mediterranean Diet, vascular stiffness, vascular structure, vascular function, vascular aging

## Abstract

Introduction: Long COVID (LC) is associated with an increase in cardiovascular risk and chronic inflammation, whereas the Mediterranean Diet (MD) seems to improve the aforementioned factors. The aim of this study is to analyse the relationship between MD and its components with vascular structure, function, and aging in patients diagnosed with LC globally and by sex. Methods: This study was a cross-sectional study with 304 subjects diagnosed with LC; 207 were women and 97 men. Adherence to MD was evaluated with a validated MEDAS questionnaire, composed of 14 items. The vascular structure was assessed using carotid intima-media thickness (cIMT). Three measurements were carried out to evaluate vascular function: cardio-ankle vascular index (CAVI), brachial-ankle pulse wave velocity (baPWV), and carotid-femoral pulse wave velocity (cfPWV). Vascular aging index (VAI) was estimated. Results: The MD score was 7.80 ± 2.33, with no difference between sexes. Vascular function and aging parameter values were higher in men than in women. Use of olive oil as the principal source of fat for cooking, and consuming <1 serving of butter/day and <1 sugar-sweetened beverage/day showed >90% adherence. Logistic regression analysis displayed associations between cIMT < 0.625 and use of olive oil in the global analysis (OR = 0.148) and among men (OR = 0.120), and <2 commercial pastries/week in global (OR = 0.536). cfPWV < 7.400 m/s was associated with DM score ≥ 8 in global (OR = 0.444) and in women, as well as with <2 pastries/week in women (OR = 0.405). baPWV < 12.315 m/s was associated with ≥3 servings of pulses/week in global (OR = 0.481) and among women, as was <2 pastries/week in global (OR = 0.471) and in women. CAVI < 7.450 was associated with ≥4 tablespoons of olive oil/day in men. VAI < 63.693 was associated with DM score ≥ 8 in global (OR = 0.458) and in women, as well as <2 pastries/week in global (OR = 0.392). Conclusions: Adherence to MD was associated with lower cfPWV and VAI measures in the global analysis and among women. In particular, several of the components were associated with a better vascular profile in LC patients.

## 1. Introduction

After levels of COVID-19 acute infection decreased, the scientific focus shifted to long COVID (LC) patients, with an estimated accumulative incidence of 400 million people [1]. LC is a disease with a wide range of manifestations, including fatigue, muscular weakness, cognitive impairment, gastrointestinal alterations, and shortness of breath. It can last for months or even longer, regardless of the severity of the initial disease [2,3]. Although the physiopathology mechanisms of LC are not well elucidated, several are probably involved, such as viral persistency, immunity and complement dysregulation, prothrombotic inflammation, or mitochondrial dysfunction [1,4,5,6,7,8,9]. In particular, microvascular endotheliopathy with low-grade inflammation is found in all these mechanisms, so it could be the genesis of a multitude of LC symptoms [10,11,12,13,14]. Moreover, endothelial dysfunction produces an increase in reactive oxygen species and vascular remodelling, boosting the activation of inflammatory cells. These molecular changes lead to several types of multi-organ damage, such as stroke, thrombosis and myocardial, and lung and liver injuries [10,12]. Therefore, it is important to ensure an in-depth and comprehensive understanding of the subject that can allow the development of therapeutical strategies to enhance endothelial function.

The Mediterranean Diet (MD) is distinguished by a high consumption of fruits, vegetables, whole grain cereals, seafood, and healthy fat like olive oil, and low consumption of red meat and processed foods. It has been demonstrated to have a beneficial effect on health by decreasing blood pressure and LDL cholesterol and improving endothelial function [15]. This reduces cardiovascular mortality rate and enhances vascular biomarkers by decreasing inflammation indicators and atheromatous plaque susceptibility. It plays a crucial role in atherosclerosis risk modulation by influencing, in metabolic ways, systemic inflammation, oxidative stress, and decrease in arterial stiffness [2,16].

Olive oil is a key component of MD, and is rich in monounsaturated fats and bioactive components, including polyphenols like oleocanthal and hydroxytyrosol [17]. They are powerful antioxidants which produce anti-inflammatory, cardioprotective and neuroprotective effects [18]. Good adherence to MD has been demonstrated to correlate with a reduced probability of contracting COVID-19 acute infection [2].

Arterial structure and arterial stiffness (AS) predict cardiovascular disease risk [19,20]. In fact, AS is associated with vascular aging (VA) [21]. Research findings have demonstrated that SARS-CoV-2 infection increases AS in the most seriously ill patients [22,23]. It is also known that COVID-19 causes early AS and VA [24]. Recent studies reveal association between AS and LC, showing that women with LC have worse arterial elasticity than men with LC [25].

To sum up, the link between AS and LC is the existence of endotheliophathy produced by an inflammation increase [26]. On the other hand, the components of MD are foods rich in polyphenols and antioxidants which have anti-inflammatory effects, modulate cell-mediated immunity, reduce inflammatory cytokine expression, and improve endothelial function [27]. However, the effects of MD on vascular structure, function, and aging in LC patients have not been studied enough, so further research is needed.

In light of the aforementioned factors, this paper aims to analyse the relationship between MD and its components with vascular structure, function, and aging in subjects diagnosed with LC in global terms (both sexes included), and by sex.

## 2. Materials and Methods

### 2.1. Design

The present study is a cross-sectional investigation, and the data were collected from subjects from the BioICOPER study. The work was conducted at the Primary Care Research Unit of Salamanca (APISAL). The protocol was published in 2023 [28]. This project was registered at Clinical Trials.gov in April 2023 (register number: NCT05819840).

### 2.2. Population

Data from 304 subjects diagnosed with long COVID (LC) were analysed, obtained from primary care records and the LC consultation data of the Internal Medicine Department at Salamanca University Hospital. The diagnostic criteria for LC defined by the World Health Organization (WHO) were utilised as inclusion criteria [29]: subjects are considered to have LC if they have symptoms for more than 2 months, which appear within 3 months of COVID-19 infection and cannot be explained by any other cause. Symptoms may persist from the acute phase, reappear after initial improvement, or fluctuate during the course of the disease [29]. Exclusion criteria were subjects being unable to attend APISAL due to their health status, or having established cardiovascular disease, or an estimated glomerular filtration rate of less than 30 mL/min/1.73 m^2^ [28]. The flow chart (Figure 1) details the subjects who participated in this study, as well as the causes for exclusion.

### 2.3. Variables and Measurement Instruments

Four trained healthcare professionals conducted tests, examinations, and analysis of the variables, following a standardised protocol [28]. The information was recorded in the REDCAP system and data quality control was performed by an independent researcher.

#### 2.3.1. Sociodemographic Variables

When participants were included in the study, their age and sex were recorded. The date of COVID-19 acute infection diagnosis was also collected, which was used to calculate the disease progression time [28].

#### 2.3.2. Adherence to the Mediterranean Diet

Adherence to the Mediterranean Diet (MD) was evaluated with a 14-item questionnaire, validated in Spain and used in the PREDIMED study [30]. The questionnaire comprises 12 questions related to the regularity of food consumption and two questions on customary eating habits for the Spanish population. Evaluation of each question was conducted using a scale of zero to one. One point was awarded for utilisation of olive oil as the primary cooking medium; daily consumption of a minimum of four tablespoons (tablespoon = 13.5 g) of olive oil, including oil used for frying, salad dressing, etc.; two or more servings of vegetables; three or more pieces of fruit; less than one cup (one cup = 100 mL) of carbonated or sugar-sweetened drinks; and consumption of white meat to a greater extent than red meat. One point was allocated for a weekly consumption of a minimum of seven glasses of wine, three or more portions of pulses, three or more servings of fish, three or more portions of nuts or dried fruits, two or more servings of sofrito (a traditional sauce made with tomato, garlic, onion, or leeks and sautéed in olive oil), and less than two commercial pastries. The range of the final score was from 0 to 14 points, where adherence to the MD was defined as ≥8 points (median value) [30].

#### 2.3.3. Vascular Structure, Vascular Function, and Vascular Aging

##### Vascular Structure

Vascular structure was assessed by carotid intima-media thickness (cIMT), employing a Sonosite Micromax ultrasound scanner (FUJIFILM Sonosite Washington, WA, USA) in accordance with a protocol previously published by our group [28].

##### Vascular Function

Arterial stiffness (AS) was analysed by measuring carotid-femoral pulse wave velocity (cfPWV), brachial-ankle pulse wave velocity (baPWV), and cardio-ankle vascular index (CAVI).

cfPWV was assessed with the patient lying down by means of SphygmoCor device (AtCor Medical Pty Ltd., West Ryde, Australia). The estimated time was determined by analysing the interval between the pulse waves of the carotid, radial, and femoral arteries relative to the R wave of the electrocardiogram (ECG). The distance was measured with a tape measure from the sternal notch to the setting of the sensor on the carotid and radial or carotid and femoral arteries [31,32].

CAVI and baPWV were assessed using the VaSera VS-2000 device (Fukuda Denshi Co., Ltd., Tokyo, Japan), connecting electrodes to the arms and ankles and a heart sound microphone to the second intercostal space, with the participant remaining silent and still. CAVI was estimated utilising the following formula: stiffness parameter β = 2ρ × 1/(SBP − DBP) × ln(SBP/DBP) × PWV, where ρ means blood density. baPWV was assessed between the aortic valve and the ankle, assuming the result obtained after three successive heartbeats to be valid [33]. The estimation was derived using the following equation: baPWV = (0.5934 × height (cm) + 14.4724)/5ba, where tba is the time elapsed between the capture of the brachial waves and the ankle waves [34].

##### Vascular Aging

The Vascular Aging Index (VAI) was estimated with the following formula [35]: VAI = (log (1.09) × 10 cIMT + log (1.14) × aPWV) × 39.1 + 4.76, where cIMT is the carotid intima-media thickness, aPWV corresponds to the aortic pulse wave velocity, which is equivalent to cfPWV, and log is the natural logarithm with base e. cIMT, a measurement of vascular structure, reveals established atherosclerosis, whilst aPWV is a metric used to assess arterial stiffness. Therefore, VAI becomes a parameter that blends measurement of different arterial properties [36].

#### 2.3.4. Lifestyles

*Tobacco use* was evaluated through the standard questionnaire adapted from the WHO MONICA study [37] and participants were categorised as either current smokers or non-smokers, if they had never smoked or had not smoked in the past year.

*Alcohol consumption* was assessed using a structured questionnaire on alcohol consumption in the preceding week, estimating the grams consumed per week.

*Physical activity* was objectively evaluated using a validated digital pedometer (Omron Hj-321 lay-UPS) [38]. This device monitors total steps, aerobic steps, distance covered in kilometers, and calories expended over the past seven days. Participants wore the pedometer for 9 days to collect data on activity over 7 full days. The average number of daily steps over a 7-day period was used as a metric to assess physical activity.

#### 2.3.5. Cardiovascular Risk Factors

##### Blood Pressure Measurement

Systolic (SBP) and diastolic (DBP) blood pressure were measured following the recommendations of the European Society of Hypertension (ESH) [36]. The measurements were taken on the participant’s dominant arm using an OMRON M10-IT sphygmomanometer (Omron Healthcare, Kyoto, Japan); the average of the last two measurements was recorded [36].

##### Analytical Parameters

A fasting sample was taken to determine the values of the following analytical parameters: total cholesterol, low-density lipoprotein cholesterol (LDL), high-density lipoprotein cholesterol (HDL), triglycerides, and fasting plasma glucose (FPG).

##### Anthropometric Measurements

Weight in kilograms was determined employing the InBody 230 monitor (InBody Co., Ltd., Seoul, Republic of Korea). Height (cm) was assessed whilst the subject was not wearing any shoes, using a height meter (Seca 222, Medical Scale and Measurement Systems, Birmingham, UK). Body mass index (BMI) was calculated by dividing weight (kg) by height (m) squared (m^2^). Obesity was defined as BMI ≥ 30. Waist circumference was measured with a tape measure parallel to the floor, above the iliac crests, at the end of exhalation with the patient standing. Abdominal obesity was considered as a waist circumference ≥ 88 cm in women and ≥102 cm in men.

Following the recommendations of the international consensus of the National Cholesterol Panel Education Programme Adult Treatment [39], participants were designated as having metabolic syndrome (MetS) if they had ≥2 of the 5 components.

### 2.4. Statistical Analysis

The mean values of the quantitative variables were compared with two categories by utilising Student’s *t*-test. The Chi-square test was employed to compare categorical variables. Logistic regression models were used to analyse the association between MD and its components with the parameters of vascular structure, function, and vascular aging. The dependent variables were cIMT, cfPWV, baPWV, CAVI, and VAI, classified into two categories using the median value (encoded as 0 = value less than the median; 1 = value greater than the median). The independent variables were adherence to MD and compliance or non-compliance with each of its components (encoded as 0 = value less than the median; 1 = value greater than the median). Regarding the adjustment variables, two models were utilised. The first model was used in the global analyses, which included age in years, and encoded sex (0 = male and 1 = female); duration of LC in months; use of antihypertensive, lipid-lowering, and/or antidiabetic drugs, coded as (0 = no, 1 = yes); cardiovascular risk factors that define MetS (encoded as the number of MetS components); and the following lifestyle factors: alcohol consumption in grams, average steps per day, and tobacco, coded as (0 = no, 1 = yes). The second model was used for the sex-specific analysis and contained the same adjustment variables, with the exception of sex. SPSS for Windows, v28.0 (IBM Corp, Armonk, NY, USA) was used for the analyses, and the cut-off point for statistical significance was *p* < 0.05.

### 2.5. Ethical Principles

The study was approved on 27 June 2022 (CEIm reference code: Ref. PI 2022 06 1048) by the “Ethics Committee for Research with medicines in the Salamanca Health Area”. The standards of good practice in observational studies established by the Declaration of Helsinki [40] and by the WHO were followed throughout the study. The confidentiality of participants was always guaranteed in accordance with Organic Law 3/2018, European Regulation 2016/679, and Council Directive 27/04/2016 on Data Protection. All participants filled out an informed consent form before being recruited for the study, after receiving information about the procedures that would be performed on them.

## 3. Results

### 3.1. Participants Characteristics

Table 1 shows the general features of the participants included in this study, both global and by sex. More women than men were included in the study (207 versus 97). Men had a higher mean age than women (55.70 ± 12.28 years versus 51.32 ± 11.54 years). Men had higher alcohol consumption, blood pressure, glucose in blood, triglycerides, BMI, and waste circumference, and a higher percentage of diagnoses of MetS. Women demonstrated higher HDL cholesterol levels than men. The time span from the beginning of SARS-CoV-2 acute infection diagnosis until study inclusion was 38.66 ± 9.58 months, with no differences between the sexes.

Table 2 shows the mean values for vascular structure, function, and aging in the global analysis and by sex. Vascular structure and aging values were higher in men than in women.

Table 3 presents a comparative analysis of the variables examined in relation to adherence to the MD. Participants with high adherence to MD were older and had higher DBP and lower triglycerides.

Table 4 shows the adherence to each MD component in the global analysis and by sex. Three components demonstrated a compliance percentage higher than 90%: use of olive oil as the principal source of fat for cooking; consumption of <1 serving of butter, margarine, or cream per day; and consumption of <1 carbonated and/or sugar-sweetened beverage per day. Men had a higher compliance percentage for weekly consumption of ≥7 cups of wine (100 mL); ≥3 portions of pulses; and ≥2 portions of boiled vegetables, pasta, rice, or other dishes with a sauce of tomato, garlic, onion, or leeks sautéed in olive oil. Women had a higher compliance percentage for intake of ≥2 servings of vegetables per day and eating white meat instead of red meat.

Figure 2 shows the compliance percentages for the 14 items of the MD adherence questionnaire according to the degree of adherence. All components had a higher percentage in the high-adherence-to-MD group (*p* < 0.005).

### 3.2. Analysis of the Association Between MD and Vascular Structure, Function, and Aging

Figure 3 reveals global OR results for both sexes concerning MD score and its components in relation to vascular structure, assessed with cIMT; the parameters of vascular function evaluated with cfPWV, baPWV and CAVI; and VAI. Figure 4 and Figure 5 show these results separated by sex, with male and female results displayed separately. MD score and 14 MEDAS components were used as independent variables. cIMT, cfPWV, baPWV, CAVI, and VAI were used as dependent variables, with the results displayed in the subplots corresponding to each variable in the figures: (a) cIMT, (b) cfPWV, (c) baPWV, (d) CAVI, and (e) VAI. Adjustment variables were age; sex; LC development time; antihypertensive, antidiabetic, and/or hypolipidemic drugs; and cardiovascular risk factors that define MetS and lifestyles: alcohol consumption (g), mean steps per day, and tobacco use.

As we can see in Figure 3a and Figure 4a, an association was found between cIMT < 0.625 mm and using olive oil as the main cooking fat in both global (OR = 0.148; 95%CI 0.043–0.509) and men (OR = 0.120; 95%CI 0.021–0.697). In addition, an association was identified between <2 commercial pastries consumption per week and cIMT < 0.625 mm in the global analysis (OR = 0.536; 95%CI 0.294–0.978).

Referring to Figure 3b and Figure 5b, cfPWV < 7.400 m/s was associated with a MD score ≥ 8 in global (OR = 0.444; 95%CI 0.253–0.780) and women (OR = 0.363; 95%CI 0.181–0.726). Likewise, cfPWV < 7.400 m/s was associated with consumption of <2 commercial pastries per week in women (OR = 0.405; 95%CI 0.198–0.829).

Furthermore, Figure 3c and Figure 5c show a correlation between baPWV < 12.315 m/s and ≥3 servings of pulses per week in both global (OR = 0.481; 95%CI 0.250–0.940), and women (OR = 0.428; 95%CI 0.187–0.977). In addition, baPWV < 12.315 m/s was associated with <2 commercial pastries consumption per week in global (OR = 0.471; 95%CI 0.246–0.902), as well as among women (OR = 0.322; 95%CI 0.140–0.742).

CAVI < 7.450 showed association with ≥4 tablespoons of olive oil per day in men (OR = 0.231; 95%CI 0.047–0.970), which can be observed in Figure 4d).

Finally, Figure 3e and Figure 5e display an association between VAI < 63.693 and MD ≥ 8 in global (OR = 0.458; 95%CI 0.243–0.863) and among women (OR = 0.243; 95%CI 0.107–0.550). Additionally, VAI < 63.693 was associated with <2 commercial pastries consumption per week in global (OR = 0.392; 95%CI 0.308–0.972).

## 4. Discussion

This study shows an association between higher adherence globally and to some of the components of the Mediterranean Diet (MD) with various parameters related to vascular structure, function, and vascular aging in patients diagnosed with long COVID (LC), especially in women. In detail, this study has revealed that adherence to MD, with a MD score ≥ 8, is associated with a favourable enhancement in cfPWV and VAI parameters. Additionally, some components of MD, such as consumption of <2 commercial pastries per week or a greater consumption of olive oil as the principal cooking medium, were associated with better cIMT values and thereby improved vascular structure. Intake of commercial pastries as well as intake of >3 pulses per week were also associated with better baPWV values both globally and in women, enhancing vascular function. These findings reinforce the hypothesis that a healthy diet, rich in anti-inflammatory and antioxidant compounds, may play a key role in preventing the vascular deterioration associated with chronic inflammatory processes such as LC.

### 4.1. Relationship Between the MD and Vascular Parameters

The results of this study showed that a higher adherence to the MD (score ≥ 8) was associated with lower carotid-femoral pulse wave velocity (cfPWV) and lower Vascular Aging Index (VAI), both in the global analysis and among women, which could indicate a higher sensitivity of the female vascular system to the benefits of diet in people with LC. We also found an OR < 1 with the other vascular parameters analysed, although these differences were not statistically significant. These findings are consistent with those published in cross-sectional studies by other authors in both the general population and in populations with cardiovascular risk factors.

Several authors have shown a negative association between adherence to the MD and vascular function parameters such as cfPWV and baPWV [22,41]. Regression analysis was used to evaluate the association between different parameters of vascular structure and arterial stiffness (AS) and adherence to MD, finding negative associations; however, it should be noted that in several cases, these associations did not reach statistical significance.

Regarding longitudinal studies, some studies have analysed patterns between adherence to MD in different populations (Spanish [42] and Dutch [43] populations) and parameters of vascular function and AS. All these studies were compiled in a recent meta-analysis published in 2025 [27], and they revealed that people adhering to a MD plan may exhibit reduced AS and, therefore, a diminished cardiovascular risk. This systematic review demonstrated an association between adherence to a MD pattern, evaluated using validated adherence scores or scales, and central AS (assessed with cfPWV), peripheral AS (evaluated with baPWV), and central and peripheral AS (measured with CAVI). It also showed an inverse relationship, which remained constant across all the included studies regardless of the outcome measure used to assess AS [27]. All these studies suggest that the beneficial effects of MD on AS are the result of a combination of mechanisms, including anti-inflammatory effects, antioxidant properties, improved lipid profiles, blood pressure regulation, and endothelial function [27].

Likewise, VAI, a vascular aging indicator that integrates cIMT and cfPWV, showed a negative association with adherence to the MD, particularly in women. These results are in line with previous publications that have analysed this relationship in subjects from populations without previous diseases [44,45]. These studies showed that subjects classified as having healthy vascular aging were more likely to adhere to the MD than subjects classified as having accelerated vascular aging. This reinforces the idea that an anti-inflammatory diet can mitigate the process of accelerated vascular aging that has been observed in individuals with LC [2].

### 4.2. Sex Differences in the Association Between Mediterranean Diet and Vascular Parameters

A relevant finding of this study is the identification of differences in association according to sex. In women, the associations between adherence to the MD and vascular markers were more consistent and significant. In men, only some individual components of the MD showed relevant relationships (e.g., use of olive oil with lower cIMT or ≥4 tablespoons of olive oil per day with lower CAVI). These findings could be related to the greater vascular vulnerability observed in women after COVID-19 in other studies, characterised by lower arterial elasticity and a more altered vascular response. Some studies have shown that women who had survived COVID-19 presented higher values in AS and inflammatory parameters, as well as higher markers of inflammation. Consequently, these results were associated with greater AS [7,8,46]. In addition, hormonal and non-hormonal factors affect gender differences. It is well established that endogenous oestrogens provide a protective effect in women until the onset of menopause. In contrast, in men, arterial stiffness increases linearly from puberty onwards, indicating that women inherently have increased main AS compared to men. These effects are lessened by sex steroids during the reproductive phase of life. Further factors that may be pertinent to this issue include height, body fat distribution, and inflammatory markers [47,48]. Furthermore, the findings indicate that women with LC appear to be more affected by central arterial stiffness; considerably higher cfPWV values were measured in women with LC compared to the control group. This suggests that, within this demographic, cfPWV serves as a more precise indicator of alterations in arterial distensibility among women [49].

### 4.3. Relevance of Specific Dietary Components

A detailed analysis of the 14 components of the MD adherence questionnaire allows us to identify those with the greatest impact on vascular health, namely, the protective role of using olive oil as the main cooking fat, consuming pulses ≥ 3 times per week, and limiting the consumption of industrial pastries. Olive oil and pulses are foods rich in polyphenols, fibre, and monounsaturated fatty acids, with proven anti-inflammatory and antioxidant effects that can improve endothelial function, reduce AS, and prevent the progression of atherosclerosis [10,12]. MD is characterised by the utilisation of olive oil, especially extra virgin olive oil, as a fundamental ingredient. It is renowned for its high concentration of monounsaturated fats and bioactive compounds, such as polyphenols like oleocanthal and hydroxytyrosol. These are potent antioxidants that have been demonstrated to produce anti-inflammatory, cardioprotective, and neuroprotective actions. Research has demonstrated that olive oil’s antioxidant properties can decrease markers of oxidative stress. This enhances the bioavailability of fat-soluble vitamins and polyphenols from other elements of the MD, thereby boosting its health benefits and establishing it as a fundamental component in precision nutrition strategies designed to reduce oxidative stress and inflammation [26]. On the other hand, pulses provide soluble fibre and bioactive compounds that enhance the lipid profile and decrease blood glucose, both of which are relevant in control of metabolic syndrome, which is common in LC [50]. Given that LC is characterised by sustained inflammation and oxidative stress, these potential effects could be particularly beneficial in preventing vascular dysfunction.

### 4.4. Comparison with Other Studies in Patients with Long COVID

Most existing studies on diet and LC have focused on the effect of diet on symptoms or disease severity in the acute phase [51,52]. Alternatively, they have examined the potential of diet and nutrition to support recovery from LC symptoms by targeting inflammation and improving outcomes [2]. Therefore, the paucity of literature analysing the relationship between diet and vascular health in this population renders this study a novel contribution to the field. Among these studies, there is a report conducted in a sub-Saharan adult population that showed that central arterial stiffness measured with cfPWV was higher in adults with LC than in controls who had never been infected with SARS-CoV-2 [49]. AS is intimately linked to the aging process, vascular injury, and the function of endothelial cells. The endothelium helps to regulate blood flow and pressure by releasing substances that affect blood vessels, such as nitric oxide, prostacyclin, endothelin-1, and platelet-activating factor [53]. Furthermore, the findings of this study are consistent with those of other studies conducted on the general population or in patients with chronic diseases, where adherence to MD has shown benefits in AS and other vascular indicators [16]. However, it should be noted that the inflammatory profile of LC could modify the magnitude or direction of these associations. Consequently, it is paramount to investigate this relationship in specific populations, such as the one studied, to ensure a comprehensive understanding of its implications.

### 4.5. Strengths and Limitations of the Study

The strengths of the study include the sample size, which is larger than most previous studies, the use of validated instruments for measuring vascular structure and function, and the adjustment for multiple confounding factors in the analyses (age, sex, lifestyle, use of medication, and cardiovascular risk factors that define MetS). In addition, a validated questionnaire was used to measure adherence to the MD, which had previously been used in the PREDIMED study.

However, several limitations should also be considered. Firstly, the cross-sectional design does not allow the establishment of causal relationships between the MD and vascular parameters. Secondly, although multiple variables were adjusted, the existence of unmeasured residual confounding factors, such as socioeconomic status or sleep quality, is possible. Thirdly, the diet questionnaire is based on self-reporting, which is susceptible to memory bias. Finally, the population included comes only from Salamanca, which may prevent the generalisation of the results to other areas or healthcare contexts. Therefore, future longitudinal studies or randomised interventions could provide more robust evidence on the impact of the MD on the progression of vascular damage in patients with LC.

### 4.6. Clinical Implications and Future Lines of Research

The results of this study indicate that encouraging dietary habits like the MD could be a useful non-pharmacological approach to reduce the vascular impact of LC, especially in women. This approach is particularly attractive in primary care, where lifestyle intervention plays a pivotal role. Future research should explore the longitudinal evolution of vascular parameters in patients with LC and whether improved adherence to the MD translates into sustained clinical benefits. It would also be relevant to analyse the role of other dietary components such as probiotics or fermented foods, which could modulate post-COVID-19 intestinal dysbiosis.

## 5. Conclusions

Adherence to MD was associated with lower cfPWV and VAI measures globally and among women. In particular, several of the components were associated with a better vascular profile in LC patients. These findings suggest that MD could become a useful dietetic strategy in order to decrease the endothelial alteration associated with LC, and highlights the need for future studies to analyse the differences between the sexes.

## Figures and Tables

**Figure 1 nutrients-17-03226-f001:**
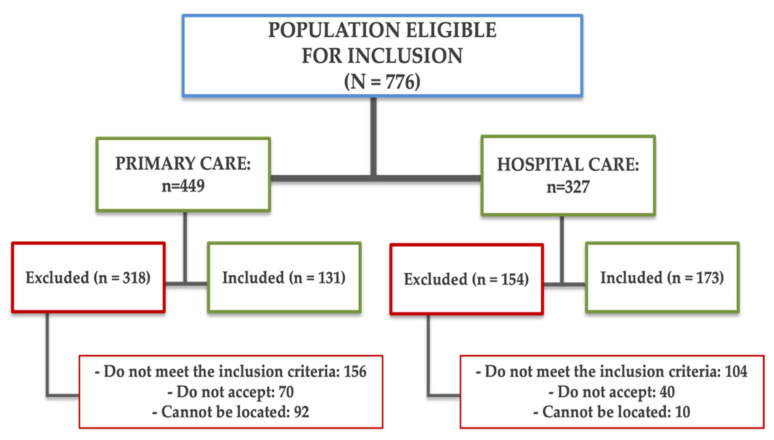
Flow chart showing the origin of the subjects included in this study, as well as the reasons for exclusion.

**Figure 2 nutrients-17-03226-f002:**
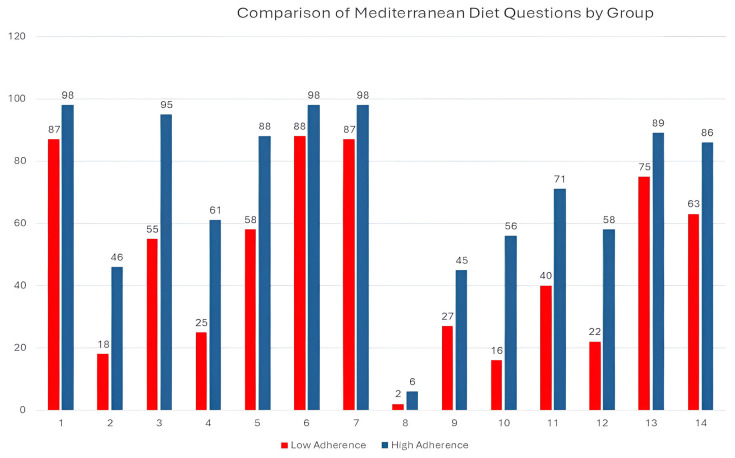
Bar chart showing the positive answer percentage for each component of the MD adherence questionnaire for both comparison groups: low adherence (<8 points, in red) and high adherence (≥8 points, in blue). 1. Do you use olive oil as the principal source of fat for cooking?; 2. Do you consume ≥4 tablespoons of olive oil per day?; 3. Do you consume ≥2 servings of vegetables per day?; 4. Do you consume ≥3 pieces of fruit per day?; 5. Do you consume <1 serving of red meat, hamburger, or sausage per day?; 6. Do you consume <1 serving of butter, margarine, or cream per day?; 7. Do you consume <1 carbonated and/or sugar-sweetened beverage per day?; 8. Do you drink ≥7 cups (100 mL) of wine per week?; 9. Do you consume ≥3 servings of pulses per week?; 10. Do you consume ≥3 servings of fish/seafood per week?; 11. Do you consume <2 commercial pastries such as cookies or cakes per week?; 12. Do you consume ≥3 servings of nuts per week?; 13. Do you prefer to eat chicken, turkey, or rabbit instead of beef, pork, hamburgers, or sausages?; 14. Do you consume boiled vegetables, pasta, rice or other dishes with a sauce of tomato, garlic, onion or leeks sautéed in olive oil ≥ 2 times per week?

**Figure 3 nutrients-17-03226-f003:**
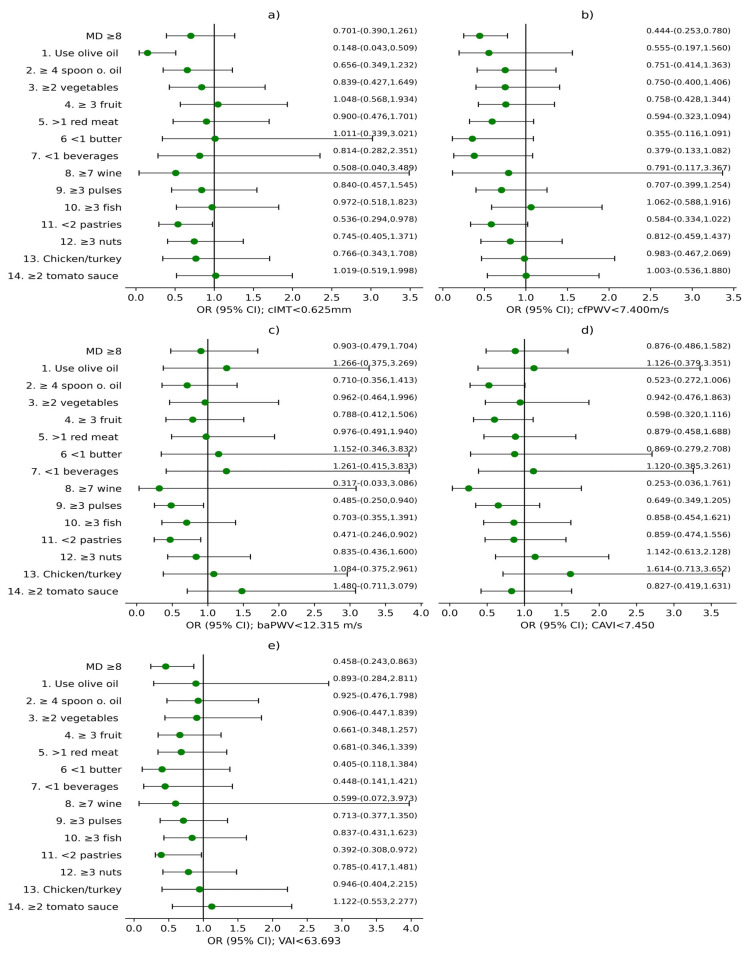
Regression logistic analysis in global analysis. (**a**) cIMT < 0.625 mm; (**b**) cfPWV < 7.400 m/s; (**c**) baPWV < 12.315 m/s; (**d**) CAVI < 7.450; (**e**) VAI < 63.693. OR: odds ratio; CI: confidence interval; cIMT: carotid intima-media thickness; cfPWV: carotid-femoral pulse wave velocity; baPWV: brachial-ankle pulse wave velocity; CAVI: cardio-ankle vascular index; VAI: Vascular Aging Index. MD ≥ 8: Mediterranean Diet score ≥ 8; 1. Do you use olive oil as the principal source of fat for cooking?; 2. Do you consume ≥4 tablespoons of olive oil per day?; 3. Do you consume ≥2 servings of vegetables per day?; 4. Do you consume ≥3 pieces of fruit per day?; 5. Do you consume <1 serving of red meat, hamburger, or sausage per day?; 6. Do you consume <1 serving of butter, margarine, or cream per day?; 7. Do you consume <1 carbonated and/or sugar-sweetened beverage per day?; 8. Do you drink ≥7 cups (100 mL) of wine per week?; 9. Do you consume ≥3 servings of pulses per week?; 10. Do you consume ≥3 servings of fish/seafood per week?; 11. Do you consume <2 commercial pastries such as cookies or cakes per week?; 12. Do you consume ≥3 servings of nuts per week?; 13. Do you prefer to eat chicken, turkey, or rabbit instead of beef, pork, hamburgers, or sausages?; 14. Do you consume boiled vegetables, pasta, rice or other dishes with a sauce of tomato, garlic, onion or leeks sautéed in olive oil ≥ 2 times per week?

**Figure 4 nutrients-17-03226-f004:**
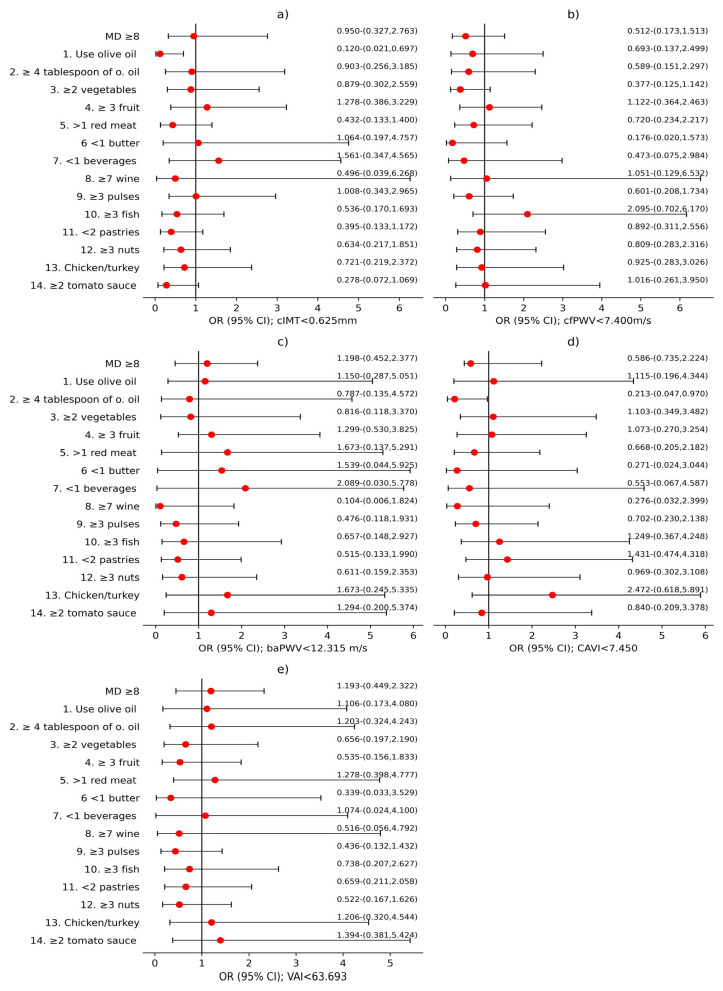
Regression logistic analysis in men. (**a**) cIMT < 0.625 mm; (**b**) cfPWV < 7.400 m/s; (**c**) baPWV < 12.315 m/s; (**d**) CAVI < 7.450; (**e**) VAI < 63.693.

**Figure 5 nutrients-17-03226-f005:**
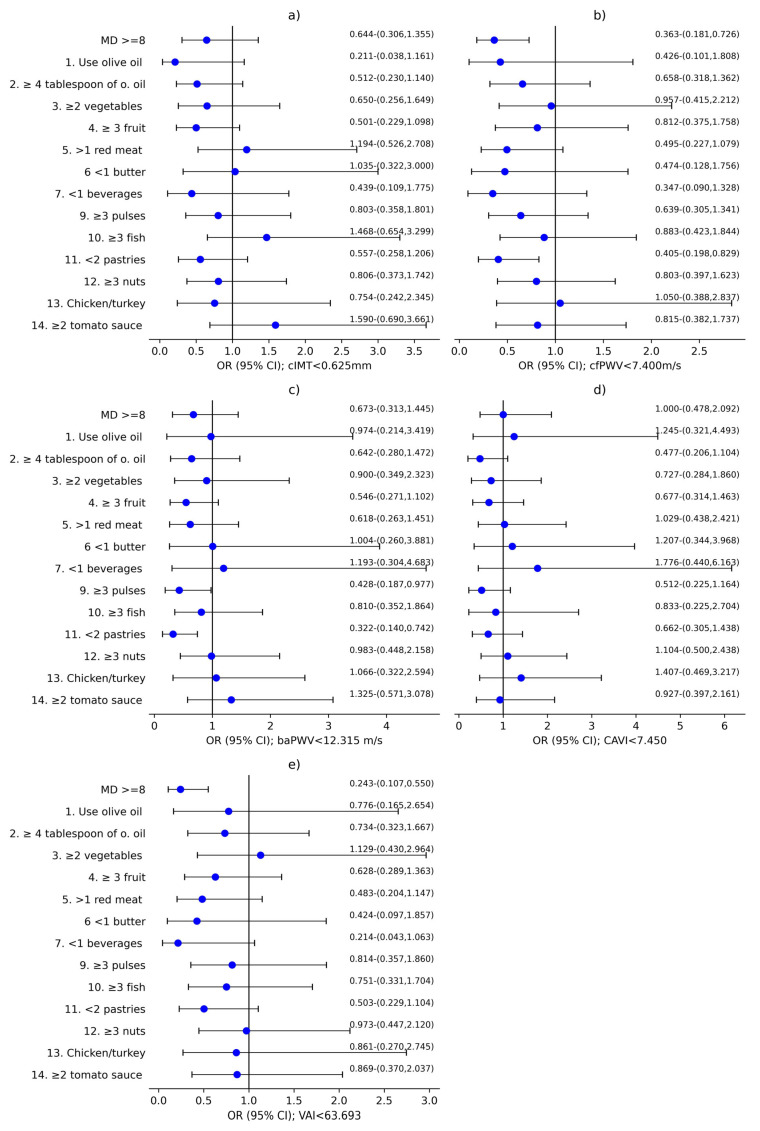
Regression logistic analysis in women. (**a**) cIMT < 0.625 mm; (**b**) cfPWV < 7.400 m/s; (**c**) baPWV < 12.315 m/s; (**d**) CAVI < 7.450; (**e**) VAI < 63.693.

**Table 1 nutrients-17-03226-t001:** Participants’ characteristics at the moment of inclusion in the global analysis and by sex.

Variable	Global (n = 304)	Men (n = 97)	Women (n = 207)	*p* Value
Age (years)	52.71 ± 11.94	55.70 ± 12.28	51.32 ± 11.54	0.001
Time with COVID-19 (months)	38.66 ± 9.58	38.74 ± 9.41	38.50 ± 9.96	0.990
**Lifestyles**				
MD (total score)	7.80 ± 2.33	7.71 ± 2.23	7.84 ± 2.39	0.487
Adherence to the MD ≥ 8, n (%)	123 (40.5)	38 (39.2)	85 (41.1)	0.427
Alcohol (g/w)	29.35 ± 52.87	60.39 ± 76.35	14.88 ± 27.19	<0.001
Steps per day	7447 ± 3633	7069 ± 3302	7624 ± 3773	0.194
**CVRF**				
Years of smoking	22.90 ±11.62	24.96 ± 12.01	21.51 ± 11.22	0.106
Active smoker, n (%)	17 (5.7)	8 (8.4)	9 (4.5)	0.065
SBP (mmHg)	119.95 ± 16.75	129.45 ± 14.37	115.52 ± 15.94	<0.001
DBP (mmHg)	76.85 ± 11.11	82.34 ± 11.04	74.30 ± 10.20	<0.001
Hypertension, n (%)	110 (36.2)	53 (54.6)	57 (27.5)	<0.001
Antihypertensives, mean ± SD	0.36 ± 0.70	0.30 ± 0.66	0.49 ± 0.78	0.024
Antihypertensives, n (%)	79 (25.9%)	45 (21.6%)	34 (35.1%)	0.013
Total cholesterol (mg/dL)	187.45 ± 34.30	182.11 ± 32.94	189.95 ± 34.71	0.029
LDL cholesterol (mg/dL)	113.03 ± 31.76	113.59 ± 32.12	112.77 ± 31.67	0.417
HDL cholesterol (mg/dL)	56.92 ± 13.58	48.78 ± 10.86	60.73 ± 13.06	<0.001
Triglycerides (mg/dL)	102.23 ± 50.81	117.47 ± 54.39	95.09 ± 47.52	<0.001
Dyslipidaemia, n (%)	172 (57.0)	67 (69.1)	105 (69.1)	0.003
Antihyperlipidemic, mean ± SD	0.29 ± 0.56	0.20 ± 0.48	0.47 ± 0.66	<0.001
Antihyperlipidemic, n (%)	75 (24.6%)	35 (16.8%)	40 (41.2%)	<0.001
FPG (mg/dL)	87.88 ± 17.67	94.37 ± 19.78	84.84 ± 15.74	<0.001
Diabetes mellitus, n (%)	37 (12.2)	22 (22.7)	15 (7.3)	<0.001
Hypoglycaemics, mean ± SD	0.11 ± 0.31	0.07 ± 0.25	0.19 ± 0.39	0.004
Hypoglycaemics, n (%)	32 (10.5%)	14 (6.7%)	18 (18.6%)	0.002
Weight (kg)	75.95 ± 17.39	88.09 ± 14.95	70.29 ± 15.46	<0.001
Height (cm)	164.50 ± 8.71	172.51 ± 7.35	160.77 ± 6.52	<0.001
BMI (kg/m^2^)	27.97 ± 5.55	29.60 ± 4.64	27.21 ± 5.78	<0.001
WC (cm)	93.92 ± 15.39	104.34 ± 12.52	89.04 ± 14.31	<0.001
Obesity, n (%)	99 (32.5)	55 (26.4)	44 (45.4)	<0.001
Number of components MetS	1.53 ± 1.34	2.04 ± 1.38	1.30 ± 1.26	<0.001
MetS, n (%)	72 (23.7)	39 (40.2)	33 (15.9)	<0.001

MD: Mediterranean Diet; CVRF: Cardiovascular risk factors; SBP: systolic blood pressure; DBP: diastolic blood pressure; FPG: fasting plasma glucose; BMI: Body Mass Index; WC: waist circumference; MetS: Metabolic Syndrome.

**Table 2 nutrients-17-03226-t002:** Comparison of categorised variables by MD adherence score (≥8 points and <8 points).

Variable	Global (n = 304)	Men (n = 97)	Women (n = 207)	*p* Value
cIMT, mean ± SD mm	0.638 ± 0.092	0.679 ± 0.116	0.619 ± 0.072	<0.001
cfPWV, mean ± SD m/s	7.797 ± 2.548	7.895 ± 1.360	7.307 ± 2.055	<0.001
baPWV, mean ± SD m/s	7.509 ± 1.261	8.854 ± 3.134	7.328 ± 1.171	<0.001
CAVI, mean ± SD	12.794 ± 2.380	13.632 ± 2.403	12.396 ± 2.268	<0.001
VAI, mean ± SD	30.871 ± 19.417	72.921 ± 17.607	63.066 ± 11.867	<0.001
cIMT ≤ 0.625, n (%)	152 (50.0)	33 (34.0)	119 (57.5)	<0.001
cfPWV ≤ 7.410, n (%),	156 (51.3)	36 (37.1)	120 (58.0)	0.001
baPWV ≤ 12.315, n (%),	151 (49.7)	30 (30.9)	121 (58.5)	<0.001
CAVI ≤ 7.400, n (%)	151 (48,7)	39 (40.2)	112 (54.1)	0.016
VAI ≤ 63.692, n (%)	151 (49.7)	29 (29.9)	122 (58.9)	<0.001

cIMT: carotid intima-media thickness; cfPWV: carotid-femoral pulse wave velocity; baPWV: brachial-ankle pulse wave velocity; CAVI: cardio-ankle vascular index; VAI: Vascular Aging Index; *p* value difference between sexes.

**Table 3 nutrients-17-03226-t003:** Comparison of categorised variables by MD adherence score (≥8 points and <8 points).

Variable	Low-Adherence MD, <8 Points (n = 181)	High-Adherence MD, ≥8 Points (n = 123)	*p* Value
Age (years)	51.71 ±12.44	54.35 ± 10.95	0.026
Time with COVID-19 (months)	29.56 ± 51.33	28.96 ± 55.42	0.462
**Lifestyles**			
Alcohol consumption (g/w)	23.94 ± 11.56	21.91 ± 11.15	0.354
Steps per day	7422 ± 3743	7483 ± 3480	0.443
**CVRF**			
Years of smoking	23.94 ± 11.56	21.91 ± 11.15	0.341
SBP (mmHg)	118.52 ± 16.14	122.11 ± 17.50	0.171
DBP (mmHg)	76.67 ± 11.20	77.19 ± 11.03	0.036 *
Total cholesterol (mg/dL)	188.49 ± 33.96	185.86 ± 35.00	0.346
LDL cholesterol (mg/dL)	113.93 ± 32.31	111.86 ± 31.12	0.258
HDL cholesterol (mg/dL)	55.81 ± 13.14	58.36 ± 14.00	0.258
Triglycerides (mg/dL)	103.46 ± 49.35	100.54 ± 53.22	0.050 *
FPG (mg/dL)	86.83 ± 13.51	89.50 ± 22.42	0.315
Weight (kg)	77.30 ± 17.20	74.13 ± 1.00	0.191
Height (cm)	164.70 ± 8.30	164.27 ± 9.33	0.682
BMI (kg/m^2^)	28.42 ± 5.66	27.35 ± 5.36	0.096
Waist circumference (cm)	94.97 ± 15.91	92.37 ± 14.76	0.145
Number of components MetS	1.59 ± 1.37	1.46 ± 1.29	0.431
cIMT, mean ± SD (mm)	0.63 ± 0.10	0.65 ± 0.08	0.194
cfPWV, mean ± SD (m/s)	7.64 ± 2.53	8.03 ± 2.56	0.193
baPWV, mean ± SD (m/s)	7.50 ± 1.25	7.60 ± 1.28	0.322
CAVI, mean ± SD	12.66 ± 2.30	12.99 ± 2.48	0.248
VAI, mean ± SD	65.18 ± 14.51	67.57 ± 14.73	0.166

The significance threshold was set at *p* < 0.05 *. Lower adherence to MD refers to <8 points in the questionnaire and high adherence refers to ≥8 points; *p* value is the comparison of groups according to the adherence. cIMT: carotid intima-media thickness; cfPWV: carotid-femoral pulse wave velocity; baPWV: brachial-ankle pulse wave velocity; CAVI: cardio-ankle vascular index; VAI: Vascular Aging Index; *p* value difference between sexes.

**Table 4 nutrients-17-03226-t004:** Adherence to Mediterranean Diet components globally and by sex.

MD Components	Global (n %)		Men (n %)		Women (n %)		*p* Value
1. USE OF OLIVE OIL AS THE PRINCIPAL SOURCE OF FAT FOR COOKING	272	(91.6%)	85	(89.5%)	187	(92.6%)	0.247
2. ≥4 TABLESPOONS OF OLIVE OIL PER DAY	88	(29.8%)	25	(26.3%)	63	(31.5%)	0.221
3. ≥2 SERVINGS OF VEGETABLES PER DAY	212	(71.6%)	58	(61.1%)	154	(76.6%)	0.005 *
4. ≥3 PIECES OF FRUIT PER DAY	118	(39.6%)	38	(39.6%)	80	(39.6%)	0.550
5. <1 SERVING OF RED MEAT, HAMBURGER, OR SAUSAGE PER DAY	208	(70.3%)	61	(63.5%)	147	(73.5%)	0.054
6. <1 SERVING OF BUTTER, MARGARINE, OR CREAM PER DAY	271	(91.9%)	90	(94.7%)	181	(90.5%)	0.155
7. <1 CARBONATED AND/OR SUGAR-SWEETENED BEVERAGE PER DAY	272	(91.3%)	88	(91.7%)	184	(91.1%)	0.530
8. ≥7 CUPS OF WINE PER WEEK	10	(3.4%)	9	(3.1%)	1	(0.3%)	<0.001 *
9. ≥3 SERVINGS OF PULSES PER WEEK	102	(34.3%)	40	(42.1%)	62	(30.7%)	0.037 *
10. ≥3 SERVINGS OF FISH/SEAFOOD PER WEEK	96	(32.4%)	28	(29.5%)	68	(33.8%)	0.271
11. <2 COMMERCIAL PASTRIES SUCH AS COOKIES OR CAKES PER WEEK	156	(52.7%)	44	(45.8%)	112	(56.0%)	0.065
12. ≥3 SERVINGS OF NUTS PER WEEK	110	(36.9%)	39	(40.6%)	71	(35.1%)	0.215
13. PREFERRING TO EAT CHICKEN, TURKEY, OR RABBIT INSTEAD OF BEEF, PORK, HAMBURGERS, OR SAUSAGES	242	(81.2%)	68	(70.8%)	174	(86.1%)	0.002 *
14. BOILED VEGETABLES, PASTA, RICE, OR OTHER DISHES WITH A SAUCE OF TOMATO, GARLIC, ONION, OR LEEKS SAUTÉED IN OLIVE OIL ≥2 TIMES PER WEEK	214	(72.1%)	75	(78.9%)	139	(68.8%)	0.045 *

* Values are numbers and proportions for categorical data. *p* value: differences between men and women.

## Data Availability

The data supporting the findings of this study are available on ZENODO under the https://zenodo.org/records/14282873 (accessed on 14 July 2025).

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
