# Peer review of "Relationship of Mediterranean Diet and Its Components with Parameters of Structure, Vascular Function, and Vascular Aging in Subjects Diagnosed with Long COVID: BioICOPER Study"

_nutrients, 2025, doi:10.3390/nu17203226_

Round 1
Reviewer 1 Report
Comments and Suggestions for Authors
This study explores the characteristics of the Mediterranean diet in patients with long COVID and its impact on vascular function and aging. The following issues need to be improved in order to enhance the quality of the manuscript.
Abstract: The English writing of this manuscript needs improvement. For example: “Vascular structure was assessed with carotid intima-media thickness (cIMT). Vascular function was evaluated by cardio-ankle vascular index (CAVI), braquial-ankle pulse wave velocity (baPWV) and carotid-femoral pulse wave velocity (cfPWV) measures. Vascular aging index (VAI) was estimated.” Although this expression is acceptable, I feel that the sentence structure is not very engaging.
What are the harms of microvascular endothelial disease? Emphasizing its severity and wide-ranging impact would help highlight the value of this manuscript.
Section 2.3: Please specify which variables were measured in this study and which measurement instruments were used.
Section 2.6.1: Why provide a detailed description of the performance of the Sonosite Micromax? I think this is not very important.
Section 2.4 and Section 2.8: What is the difference between them?
Please use punctuation correctly. I have already found many punctuation errors in the article, such as in Line 102.
Line 312–313: Could you write out the specific trends of change? For example, delaying aging or enhancing which functions?
Discussion: The authors list many references, but I believe this is inappropriate. The literature should be discussed in combination with the authors’ own research.
I reviewed the similarity check results and I think your repetition rate is somewhat high. Please revise sentences in the Results and Discussion sections as appropriate.
Table 1: I don’t think there should be a comma after Age or in similar cases, except in items like Hypertension, n (%); Antihypertensives, n (%), etc.
Reviewer 2 Report
Comments and Suggestions for Authors
In this manuscript (ID#: nutrients-3912253), entitled “Relationship of Mediterranean Diet and Its Components with Parameters of Structure, Vascular Function, and Vascular Aging in Subjects Diagnosed with Long Covid: BioICOPER Study”, authors: Navarro-Caceres et al analyze the relationship between adherence to the Mediterranean diet and vascular structure/function in patients with Long Covid. Their results suggest that adherence to the Mediterranean diet is associated with better vascular function in these patients. They further conclude that specific dietary components are linked to improved vascular profiles. However, the experimental design lacks rigor, and the reliability of the results is questionable. Major concerns are summarized below:
- Drug use confounding: Table 1 indicates that some patients were taking antihyperlipidemic and antihypertensive medications. How were the potential confounding effects of these drugs excluded or controlled for in the analysis?
- Unclear role of dietary components: The authors conclude that certain components of the Mediterranean diet are associated with better vascular profiles. However, evidence directly supporting this claim is lacking. Which specific components were associated with these effects, and how was this determined?
- Inconsistent conclusions (Table 3): The data presented in Table 3 do not appear to support the authors’ conclusions. In particular, there are no significant differences in cfPWV or VAI between low- and high-adherence Mediterranean diet groups. These results should be carefully verified and the conclusions revised accordingly.
- Methodological details for VAI: The manuscript reports measurement of the vascular aging index (VAI), but no clear description of the measurement method is provided. Please specify the methodology used for VAI determination.
- Endothelial function not specified: The authors suggest that vascular alterations are related to endothelial dysfunction. However, no direct evidence of endothelial function was presented. Vascular changes could be influenced by multiple factors, including local inflammation, vascular smooth muscle function, blood pressure, and lipid metabolism. Please clarify how endothelial function was specifically assessed in this study.
- Minor Concerns: 1) The abstract is overly long and should be shortened to emphasize the most important findings. 2) Several terms should be carefully revised or translated to avoid misinterpretation: Estilos de vida, egliahs → translate into English; Nº de pasos día → translate into English; Global → clarified, as the current usage seems intended to mean “both genders.”
In this manuscript (ID#: nutrients-3912253), entitled “Relationship of Mediterranean Diet and Its Components with Parameters of Structure, Vascular Function, and Vascular Aging in Subjects Diagnosed with Long Covid: BioICOPER Study”, authors: Navarro-Caceres et al analyze the relationship between adherence to the Mediterranean diet and vascular structure/function in patients with Long Covid. Their results suggest that adherence to the Mediterranean diet is associated with better vascular function in these patients. They further conclude that specific dietary components are linked to improved vascular profiles. However, the experimental design lacks rigor, and the reliability of the results is questionable. Major concerns and recommendations have been provided to the authors.
Round 2
Reviewer 1 Report
Comments and Suggestions for Authors
Approved
Reviewer 2 Report
Comments and Suggestions for Authors
The revised manuscript has been improved significantly. No further recommendation.